# On the Approximation of Spatial Structures of Global Tidal Magnetic Field Models

Roger Telschow[1], Christian Gerhards[1], and Martin Rother[2]

[1]Computational Science Center, University of Vienna, 1090 Vienna, Austria
[2]Helmholtz Centre Potsdam German Research Centre for Geosciences – GFZ, Section 2.3 Geomagnetism, 14467 Potsdam, Germany

**Correspondence:** Roger Telschow (roger.telschow@univie.ac.at)

**Abstract.** The extraction of the magnetic signal induced by the oceanic M2 tide is typically based solely on the temporal periodicity of the signal. Here, we propose a system of tailored trial functions that additionally takes the spatial constraint into account that the sources of the signal are localized within the oceans. This construction requires knowledge of the underlying conductivity model but not of the inducing tidal current velocity. Approximations of existing tidal magnetic field models with these trial functions and comparisons with approximations based on other localized and non-localized trial functions are illustrated.

## 1 Introduction

Conductive sea water moving through the ambient Earth's main magnetic field $\mathbf{B}_{\mathrm{main}}$ induces a secondary magnetic field $\mathbf{B}_{\mathrm{oc}}$. Due to their periodic nature, magnetic signals generated by ocean tides are particularly easy to detect and have been studied in observatory data as early as, e.g., Malin (1970). However, the extraction of global models for magnetic fields induced by the dominating M2 tide from satellite data has become possible only recently (e.g., Sabaka et al. (2015, 2016); Tyler et al. (2003)). Although the used extraction procedures are solely based on the temporal periodicity of the tidal signal (and not on further information on the spatial localization of the sources), they seem, by visual inspection, to coincide very well with results obtained by forward models such as in Kuvshinov and Olsen (2005). A more extensive comparison of forward models of electromagnetic ocean tidal signals based on different ocean tide models has recently been published in Saynisch et al. (2018). In that work, it was shown that the residuals between the different models can exceed the nominal noise level of the Swarm satellite mission. The ability to extract M2 tidal magnetic field signals in satellite data more precisely can therefore help in constraining ocean tide models. In Grayver et al. (2016); Schnepf et al. (2015) it has been shown that an M2 tidal magnetic field model can also be used to constrain 1-D models of the Earth's mantle conductivity, and forward studies in Irrgang et al. (2016) have shown that lateral variations in the conductivity of the ocean water itself should have a detectable influence on the measured magnetic field (although the latter study was performed for general ocean circulation and not for tidal current systems).

In this short paper, we want to illustrate the effects that different (spatially localized) sets of trial functions can have on the approximation of the magnetic field induced by the M2 tide in the first place.

In particular, we describe a possible setup for the inclusion of spatial localization constraints (in addition to the constraint of temporal periodicity) for the approximation of ocean tide induced magnetic fields. Clearly, the velocity field $\mathbf{u}$ that is responsible for the generation of the corresponding secondary magnetic field $\mathbf{B}_{\mathrm{oc}}$ vanishes over the continents. The precise connection between $\mathbf{u}$ and $\mathbf{B}_{\mathrm{oc}}$ is given by the time-harmonic Maxwell equations

$$
\begin{aligned}
\nabla \times \mathbf{B}_{\mathrm{oc}} &= \mu_0 \sigma (\mathbf{E} + \mathbf{u} \times \mathbf{B}_{\mathrm{main}}), \\
\nabla \times \mathbf{E} &= i\omega \mathbf{B}_{\mathrm{oc}}, \\
\nabla \cdot \mathbf{B}_{\mathrm{oc}} &= 0,
\end{aligned} \tag{1}
$$

where we assume to have knowledge of $\mathbf{B}_{\mathrm{main}}$, the underlying conductivity $\sigma$, and the frequency $\omega$. Furthermore, $\mathbf{E}$ denotes the electric field and $\mu_0$ is the vacuum permeability. Instead of using a fixed velocity field $\mathbf{u}$, we substitute it by a set of functions $\{\mathbf{u}_\ell\}_{\ell=1,\ldots,L}$ (e.g., vectorial Slepian functions as in Plattner and Simons (2014, 2015) that are localized over the oceans) to obtain a set of corresponding trial functions $\{\mathbf{B}_\ell\}_{\ell=1,\ldots,L}$ that each solve (1). The latter is suitable for the approximation of $\mathbf{B}_{\mathrm{oc}}$ and reflects the spatial localization of the sources of the induced magnetic signal in the oceans. Thus, a magnetic field model that is based on an expansion of the signal in terms of the function system $\{\mathbf{B}_\ell\}_{\ell=1,\ldots,L}$ automatically reflects the spatial origin of the signal as well as its temporal periodicity (described by the frequency $\omega$). Additionally, due to the linear connection between $\mathbf{u}$ and $\mathbf{B}_{\mathrm{oc}}$, an approximation of $\mathbf{B}_{\mathrm{oc}}$ directly yields an approximation of the underlying tidal current velocity $\mathbf{u}$ in terms of the functions $\{\mathbf{u}_\ell\}_{\ell=1,\ldots,L}$. However, a model of the underlying conductivity $\sigma$ has to be assumed for the construction of the $\mathbf{B}_\ell$. Throughout this paper, we fix the underlying conductivity, meaning that we do not test the influence of a variation of the conductivity model on the approximation of $\mathbf{B}_{\mathrm{oc}}$. The goal of the paper is rather the illustration of the effect of the general constraint that the (unknown) underlying $\mathbf{u}$ is restricted to the oceans. In a forthcoming study, the simultaneous reconstruction of $\mathbf{u}$ and approximation of $\mathbf{B}_{\mathrm{oc}}$, and a comparison with existing models, shall be investigated more thoroughly. Since the connection between $\sigma$ and $\mathbf{B}_{\mathrm{oc}}$ is nonlinear, a simultaneous determination of $\sigma$ and $\mathbf{B}_{\mathrm{oc}}$ (assuming a fixed velocity field model for $\mathbf{u}$) is not as straightforward. A detailed description of the trial functions is provided in Section 2.

In Section 3, we illustrate our approach with input data derived from the (satellite and observatory data based) CM5 model of Sabaka et al. (2015) and from data derived from a forward model based on the X3DG solver from Kuvshinov (2008). We approximate these input data sets separately in terms of time-periodic vector spherical harmonics, a system of spatially localized trial functions that contains no particular information on the underlying sources (in our case, Abel-Poisson kernels), and the new set of trial functions indicated in the previous paragraph, respectively. We also include an example with artificial continental noise. The residuals with respect to the input data show that the use of the function system $\{\mathbf{B}_\ell\}_{\ell=1,\ldots,L}$ can filter out undesired contributions to the M2 tidal magnetic field over the continents, without neglecting data over the continents. These residuals can reach up to 15% of the maximal signal strength and have a magnitude that should be detectable at satellite altitude.

## 2 Method and Function Systems

Given a so-called dictionary $\mathcal{D}$ of trial functions, we use the Regularized (Orthogonal) Functional Matching Pursuit (cf. Fischer and Michel (2013); Michel and Telschow (2014, 2016) for details) for the approximation of $\mathbf{B}_{\mathrm{oc}}$. Shortly speaking, this is a greedy-type algorithm that yields an approximation

$$5 \quad \bar{\mathbf{B}}_N = \sum_{i=1}^{N} \alpha_i \mathbf{d}_i$$

of $\mathbf{B}_{\mathrm{oc}}$ by iteratively choosing coefficients $\alpha_i \in \mathbb{R}$ and dictionary elements $\mathbf{d}_i \in \mathcal{D}$ via

$$\underset{\alpha \in \mathbb{R}, \mathbf{d} \in \mathcal{D}}{\operatorname{argmin}} \left( \|\mathbf{R}_{i-1} - \alpha \mathcal{F}\mathbf{d}\|_{\mathbb{R}^M}^2 + \lambda \|\bar{\mathbf{B}}_{i-1} + \alpha \mathbf{d}\|_{\mathcal{H}}^2 \right). \tag{2}$$

$\mathbf{R}_{i-1} = \mathbf{b} - \mathcal{F}\bar{\mathbf{B}}_{i-1}$ denotes the residual between the data $\mathbf{b} \in \mathbb{R}^M$ and the approximation after $i-1$ iterations. In this particular setup, $\mathcal{F}$ represents the linear operator that evaluates a function at the $M$ locations where data is provided, and $\mathcal{H}$ is a suitable

Hilbert space for the regularization of the problem[1]. The parameter $\lambda$ controls the trade-off between the data misfit $\|\mathbf{R}_i\|_{\mathbb{R}^M}^2$ and the regularizing term $\|\bar{\mathbf{B}}_i\|_{\mathcal{H}}^2$, which imposes a certain porperty to $\bar{\mathbf{B}}_i$ such as smoothness (as in our case). The Regularized (Orthogonal) Functional Matching Pursuit, in general, has the advantage that it can easily deal with different dictionaries $\mathcal{D}$ (or combinations of such) from which the approximant $\bar{\mathbf{B}}_N$ is built. However, any other approximation method could be used as well with the proposed function systems. In this paper, we use the term "dictionary" simply to describe a set of arbitrary

functions that we consider suitable for our purposes. These functions do not necessarily have to satisfy particular mathematical properties such as orthogonality or completeness. Therefore, we call such functions "trial functions" rather than, e.g., "basis functions".

In the following, we briefly introduce some function systems that can be used for the constitution of $\mathcal{D}$. In particular, Section 2.4 describes in more detail the already mentioned trial functions $\{\mathbf{B}_\ell\}_{\ell=1,\ldots,L}$ that contain temporal and spatial constraints

tailored for ocean tide induced magnetic fields.

### 2.1 Vector Spherical Harmonics

We briefly recapitulate the notion of classical vector spherical harmonics in a form that we need at a few occasions later on. By $\mathbb{S}_r = \{x \in \mathbb{R}^3 : |x| = r\}$, we denote the sphere of radius $r$, while $\mathbb{S} = \mathbb{S}_1$ stands short for the unit sphere. Every unit vector $\xi = \xi(t, \varphi) \in \mathbb{S}$ can be expressed in spherical coordinates with longitude $\varphi$ and polar distance $t = \cos(\vartheta)$, where $\vartheta$ is the

corresponding co-latitude. By $Y_{n,k}$, we denote fully normalized spherical harmonics of degree $n$ and order $k$: for every $n \in \mathbb{N}_0$ and $k = -n, \ldots, n$,

$$Y_{n,k}(\xi) = \sqrt{\frac{2n+1}{4\pi} \frac{(n-|k|)!}{(n+|k|)!}} \, P_n^{|k|}(t) \begin{cases} \sqrt{2}\cos(k\varphi), & k < 0, \\ 1, & k = 0, \\ \sqrt{2}\sin(k\varphi), & k > 0. \end{cases}$$

---

[1]In our case, we use the norm $\|f\|_{\mathcal{H}}^2 = \sum_{n,k}(n+\frac{1}{2})^4 \hat{f}_{(n,k)}$ but other norms can be used as well depending on the property one wants to impose on $f$. By $\hat{f}_{(n,k)}$ we denote the Fourier coefficients of $f$ as indicated in (3).

The involved associated Legendre functions (ALFs) are, for $t \in [-1, 1]$, defined as

$$P_n^k(t) = \frac{(-1)^k}{2^n n!} \left(1 - t^2\right)^{k/2} \left(\frac{d}{dt}\right)^{n+k} (t^2 - 1)^n.$$

If $f$ is a scalar-valued square-integrable function on $\mathbb{S}$, then, for every degree $n \in \mathbb{N}_0$ and order $k = -n, \ldots, n$, the values

$$\hat{f}_{(n,k)} = \int_{\mathbb{S}} f(\eta) Y_{n,k}(\eta) dS(\eta) \tag{3}$$

are called the Fourier coefficients of the function $f$.

Going over to the vectorial setting, it is well-known that every square-integrable vector field $\mathbf{f}$ on the unit sphere can be uniquely decomposed into its radial and two tangential components such that

$$\mathbf{f} = \mathbf{e}_r f_1 + \nabla_{\mathbb{S}} f_2 + L_{\mathbb{S}} f_3$$

with scalar-valued functions $f_1, f_2, f_3$ and the radial unit vector $\mathbf{e}_r = (\sqrt{1 - t^2} \cos(\varphi), \sqrt{1 - t^2} \sin(\varphi), t)^T$. By the surface gradient $\nabla_{\mathbb{S}}$, we denote the tangential component of the usual Euclidean gradient $\nabla$, i.e.,

$$\nabla_{\mathbb{S}} = \mathbf{e}_\varphi \frac{1}{\sqrt{1 - t^2}} \frac{\partial}{\partial \varphi} + \mathbf{e}_t \sqrt{1 - t^2} \frac{\partial}{\partial t},$$

with unit vectors $\mathbf{e}_t = (-t \cos(\varphi), -t \sin(\varphi), \sqrt{1 - t^2})^T$ and $\mathbf{e}_\varphi = (-\sin(\varphi), \cos(\varphi), 0)^T$. Moreover, the surface curl gradient $L_{\mathbb{S}}$ is defined by $L_{\mathbb{S}} f(\xi) = \xi \times \nabla_{\mathbb{S}} f(\xi)$, where $\times$ is the usual cross product in $\mathbb{R}^3$. In other words,

$$L_{\mathbb{S}} = -\mathbf{e}_\varphi \sqrt{1 - t^2} \frac{\partial}{\partial t} + \mathbf{e}_t \frac{1}{\sqrt{1 - t^2}} \frac{\partial}{\partial \varphi}.$$

Hence, we define three types of vector spherical harmonics: the radial

$$\mathbf{y}_{n,k}^{(1)} = \mathbf{e}_r Y_{n,k},$$

for degrees $n \geq 0$ and orders $k = -n, \ldots, n$, as well as the tangential

$$\mathbf{y}_{n,k}^{(2)} = \sqrt{\frac{1}{n(n+1)}} \nabla_{\mathbb{S}} Y_{n,k}, \tag{4}$$

$$\mathbf{y}_{n,k}^{(3)} = \sqrt{\frac{1}{n(n+1)}} L_{\mathbb{S}} Y_{n,k}, \tag{5}$$

for degrees $n \geq 1$ and orders $k = -n, \ldots, n$. Note that, for convenience, we set $\mathbf{y}_{0,0}^{(2)} = \mathbf{y}_{0,0}^{(3)} = \mathbf{0}$. It should further be noted that the vector spherical harmonics in (4) are surface curl-free while those in (5) are surface divergence-free. In analogy to (3), we can now define the Fourier coefficients

$$\hat{f}_{(n,k)}^{(i)} = \int_{\mathbb{S}} \mathbf{f}(\eta) \cdot \mathbf{y}_{n,k}^{(i)}(\eta) dS(\eta)$$

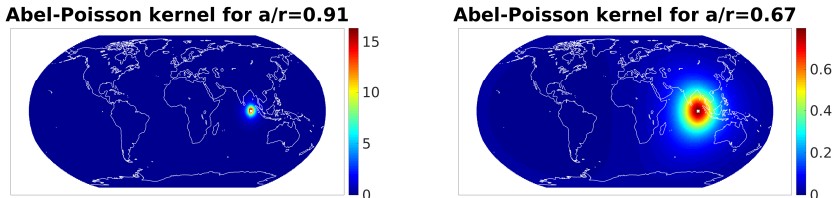

**Figure 1.** The kernel $K(r\cdot, a\eta_1)$ for $\frac{a}{r} = 0.91$ (left) and $\frac{a}{r} = 0.67$ (right). The fixed nodal point $\eta_1 \in \mathbb{S}$ is marked by a white cross.

of square-integrable vector fields $\mathbf{f}$ on the unit sphere.

The vector spherical harmonics from above are defined solely on the unit sphere and can, therefore, only be used for the expansion of vector-valued functions on $\mathbb{S}$. However, for the approximation of satellite potential field data it is necessary to have related functions that are also defined in the exterior of a sphere. For that purpose, we define the following gradients of harmonic extensions of (scalar) spherical harmonics:

$$\mathbf{h}_{n,k}(x) = \frac{1}{r^2} \left(\frac{a}{r}\right)^n \left(\nabla_{\mathbb{S}} Y_{n,k}(\xi) - \xi(n+1)Y_{n,k}(\xi)\right), \tag{6}$$

for $r = |x| > a$ and $\xi = \frac{x}{|x|} \in \mathbb{S}$, where $a$ is the radius of a reference sphere, e.g., Earth's mean radius.

### 2.2 Vectorial Abel-Poisson Kernel

While the set of functions $\{\mathbf{h}_{n,k}\}_{n \in \mathbb{N}_0, k=-n,\dots,n}$ from (6) is suitable for the global approximation of potential field data, we are also interested in localized functions. One possible choice is the Abel-Poisson kernel (see, e.g., Freeden and Gerhards (2012); Freeden et al. (1998)). For $x, y \in \mathbb{R}^3$, $|x| > |y|$, it is defined by

$$K(x,y) = \frac{1}{4\pi} \frac{|x|^2 - |y|^2}{|x-y|^3}.$$

That is, with unit vectors $\xi, \eta \in \mathbb{S}$ and radii $r > a > 0$, we have

$$K(r\xi, a\eta) = \frac{1}{4\pi} \frac{r^2 - a^2}{\left(a^2 + r^2 - 2ar(\xi \cdot \eta)\right)^{3/2}}$$

which shows that $K$ only depends on the spherical distance between $\xi$ and $\eta$, since $|\xi - \eta|^2 = 2(1 - \xi \cdot \eta)$. Therefore, the kernel is radially symmetric if we keep one of the variables fixed (we strictly keep the second argument fixed, here $a\eta$). The degree of localization is determined by the ratio $\frac{a}{r}$. The closer it is to one, i.e., the smaller the difference between the radii $a$ and $r$, the better is the spatial localization of $K(r\cdot, a\eta)$ around $\eta$. In our case, we choose $a$ to be the Earth's mean radius, and $r$ is the radius of the sphere at which we evaluate the kernel. An illustration of the kernel is provided in Figure 1.

The corresponding vectorial Abel-Poisson kernel is simply defined by

$$\mathbf{k}(x,y) = \nabla_x K(x,y) = \frac{1}{4\pi} \sum_{n=0}^{\infty} \sum_{k=-n}^{n} Y_{n,k}\left(\frac{y}{|y|}\right) \mathbf{h}_{n,k}(x).$$

Further calculations show

$$\mathbf{k}(x,y) = \frac{1}{4\pi}\left(\frac{2}{|x-y|^3}x - 3\frac{|x|^2 - |y|^2}{|x-y|^5}(x-y)\right). \tag{7}$$

A spatially localized alternative to $\{\mathbf{h}_{n,k}\}_{n\in\mathbb{N}_0, k=-n,\dots,n}$ could then be defined by the set of functions $\{\mathbf{k}(\cdot, a\eta_i)\}_{i=1,\dots,M}$, where $\eta_1,\dots,\eta_M \in \mathbb{S}$ is a fixed set of adequately distributed nodal points.

## 2.3 Spherical Slepian Functions

While the localization of Abel-Poisson kernels is of radially symmetric nature, one is often interested in regions of more complex geometry, e.g., continents or oceans. Spherical Slepian functions, for instance, provide an orthonormal system of functions that can reflect localization in such general predefined regions $\Gamma \subset \mathbb{S}$ (see, e.g., Plattner and Simons (2015, 2017a); Simons et al. (2006); Simons and Plattner (2015) for details).

Specifically, the function $\mathbf{f}$ showing the best localization in $\Gamma$, is the one that maximizes the energy ratio

$$\lambda_\Gamma(\mathbf{f}) = \frac{\int_\Gamma |\mathbf{f}(\eta)|^2 \mathrm{d}S(\eta)}{\int_\mathbb{S} |\mathbf{f}(\eta)|^2 \mathrm{d}S(\eta)}, \tag{8}$$

i.e., the one with an energy ratio closest to one. Let us now assume that $\mathbf{g}^{(i)}$ is a bandlimited vectorial function of type $i$ with bandlimit $N$, i.e., it can be expanded as

$$\mathbf{g}^{(i)} = \sum_{n=0}^{N}\sum_{k=-n}^{n} \hat{g}_{(n,k)}^{(i)}\mathbf{y}_{n,k}^{(i)}.$$

Further, the matrix $\mathbf{P} = (P_{(n,k),(m,j)}) \in \mathbb{R}^{(N+1)^2 \times (N+1)^2}$ contains (properly sorted[2]) all of the appearing inner products

$$P_{(n,k),(m,j)} = \int_\Gamma \mathbf{y}_{n,k}^{(i)}(\eta) \cdot \mathbf{y}_{m,j}^{(i)}(\eta)\mathrm{d}S(\eta)$$

and $\hat{\mathbf{g}} = (\hat{g}_{(n,k)}^{(i)})^{\mathrm{T}} \in \mathbb{R}^{(N+1)^2}$ with $n = 0,\dots,N$ and $k = -n,\dots,n$. If we now restrict ourselves to normalized functions $\mathbf{g}^{(i)}$ (i.e., $\int_\mathbb{S} |\mathbf{g}^{(i)}(\eta)|^2 \mathrm{d}S(\eta) = \hat{\mathbf{g}}^{\mathrm{T}}\hat{\mathbf{g}} = 1$), one obtains the simple expression $\lambda_\Gamma(\mathbf{g}^{(i)}) = \hat{\mathbf{g}}^{\mathrm{T}}\mathbf{P}\hat{\mathbf{g}}$. Eventually, the maximization of the energy ratio (8) leads to the eigenvalue problem

$$\mathbf{P}\hat{\mathbf{g}} = \lambda\hat{\mathbf{g}}.$$

The eigenvalues $\lambda_\ell$ are the possible energy ratios and the corresponding eigenvectors $\hat{\mathbf{g}}_\ell$ contain the Fourier coefficients of bandlimited functions $\mathbf{g}_\ell^{(i)}$ attaining the energy ratio $\lambda_\ell = \lambda_\Gamma(\mathbf{g}_\ell^{(i)})$. The set of functions $\{\mathbf{g}_\ell^{(i)}\}_{\ell=1,\dots,(N+1)^2}$ is ordered such that $1 \geq \lambda_1 \geq \lambda_2 \geq \dots \geq \lambda_{(N+1)^2} \geq 0$.

In typical scenarios, it turns out that the eigenvalues are clustered close to one and close to zero. Those eigenvalues $\lambda_1,\dots,\lambda_L$ which are closer to one determine the subset $\{\mathbf{g}_\ell^{(i)}\}_{\ell=1,\dots,L}$ of well-localized Slepian functions that should be used for approximation in $\Gamma$. The code for the generation of vectorial Slepian functions has been kindly supplied in Plattner and Simons (2017b). For our situation, where $\Gamma$ denotes the region of (a spherical) Earth which is covered by oceans, an illustration is provided in Figure 2.

---

[2]typically, the order is $(0,0),(1,-1),(1,0),(1,1),(2,-2),\dots$ such that the pair $(n,k)$ is at position $n^2 + n + k + 1$ in a row or column.

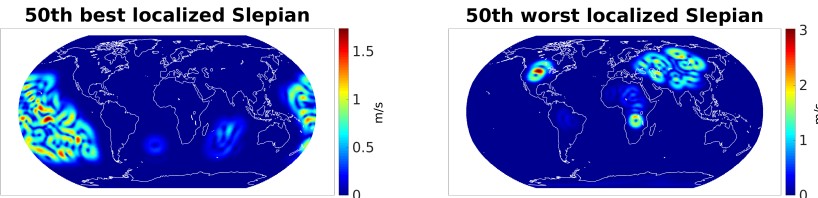

**Figure 2.** Absolute value of the vectorial Slepian function $\mathbf{g}_{50}^{(3)}$ with 50th-best localization over the oceans (left) and $\mathbf{g}_{1630}^{(3)}$ with the 50th-worst localization over the oceans (right), for bandlimit $N = 40$.

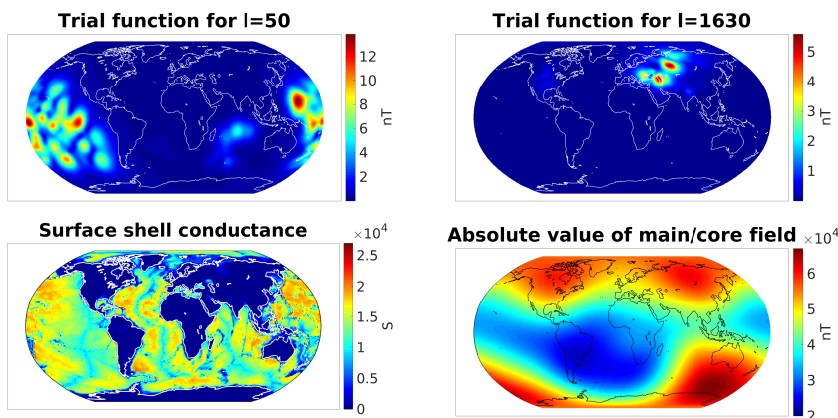

**Figure 3.** Absolute value of the trial function $\mathbf{B}_{50}^{\mathrm{re}}$ corresponding to $\mathbf{u} = \mathbf{g}_{50}^{(3)}$ at time $t = 0$ (top left) and the trial function $\mathbf{B}_{1630}^{\mathrm{re}}$ corresponding to $\mathbf{u} = \mathbf{g}_{1630}^{(3)}$ at time $t = 0$ (top right). Bottom row: Models of surface shell conductance on $\mathbb{S}_a$ and absolute value of $\mathbf{B}_{\mathrm{main}}$ on $\mathbb{S}_a$ used for the generation of the trial functions.

## 2.4   Physics Based Trial Functions

We start with the time-harmonic Maxwell equations as already indicated in (1). For simplicity, we assume a 1-D (only radially varying) conductivity model for $\sigma$ within the ball $\mathbb{B}_a$ and at the surface $\mathbb{S}_a$ we allow a laterally varying conductivity (cf. the bottom left image in Figure 3 for an illustration). Further, the magnetic field $\mathbf{B}_{\mathrm{main}}$ is taken from the CHAOS-5 model (see Finlay et al. (2015)) and $\mathbf{u}$ is supposed to denote a depth-integrated velocity field that is restricted to $\mathbb{S}_a$ (in fact, within the numerical framework of the X3DG solver, we assume a constant ocean depth of 1km, with $\mathbf{u}$ being tangential to the sphere and independent of the depth). Since we are mainly interested in tidal velocity fields, it is a reasonable assumption that $\mathbf{u}$ is surface divergence-free for most parts of the oceans. The latter means that $\mathbf{u}$ can be expanded in terms of vector spherical harmonics or vectorial Slepian functions of type 3, i.e., $\mathbf{y}_{n,k}^{(3)}$ or $\mathbf{g}_{\ell}^{(3)}$, respectively.

For the generation of the tailored trial functions $\{\mathbf{B}_\ell\}_{\ell=1,\dots,L}$, we, therefore, substitute $\mathbf{u}$ by a set of surface divergence-free functions $\{\mathbf{u}_\ell\}_{\ell=1,\dots,L}$ that reflect spatial localization within the oceans. More precisely, we choose

$$\mathbf{u}_\ell = \mathbf{g}_\ell^{(3)},$$

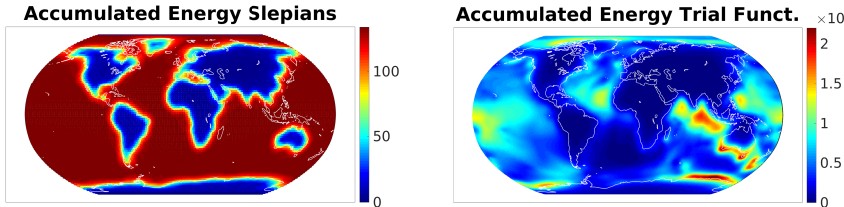

**Figure 4.** Accumulated energy for $\{\mathbf{u}_\ell\}_{\ell=1,\ldots,L}$ (left) and $\{\mathbf{B}_\ell^{\mathrm{re}}, \mathbf{B}_\ell^{\mathrm{im}}\}_{\ell=1,\ldots,L}$ (right), with $L = 1200$.

where $\mathbf{g}_\ell^{(3)}$ is the $\ell$-th best localized vectorial Slepian function of type 3. The corresponding solution $\mathbf{B}_{\mathrm{oc}}$ of (1) within this setup then provides an auxiliary function $\tilde{\mathbf{B}}_\ell$. It should be noted that in order to obtain Maxwell's equation in the time-harmonic form (1), one has to apply a Fourier transform in time. Therefore, for the actual trial function $\mathbf{B}_\ell$, we have to invert the Fourier transform and get

$$\mathbf{B}_\ell(x,t) = e^{-i\omega t}\tilde{\mathbf{B}}_\ell(x), \quad x \in \mathbb{R}^3, t \in \mathbb{R}.$$

For technical reasons, we choose to work in a real-valued framework, so that the real and imaginary part of $\mathbf{B}_\ell$ each yield a trial function

$$\mathbf{B}_\ell^{\mathrm{re}}(x,t) = \cos(\omega t)\tilde{\mathbf{B}}_\ell^{\mathrm{re}}(x) + \sin(\omega t)\tilde{\mathbf{B}}_\ell^{\mathrm{im}}(x), \tag{9}$$

$$\mathbf{B}_\ell^{\mathrm{im}}(x,t) = \sin(\omega t)\tilde{\mathbf{B}}_\ell^{\mathrm{re}}(x) - \cos(\omega t)\tilde{\mathbf{B}}_\ell^{\mathrm{im}}(x). \tag{10}$$

Thus, each choice of $\mathbf{u}_\ell$ yields two functions $\mathbf{B}_\ell^{\mathrm{re}}$ and $\mathbf{B}_\ell^{\mathrm{im}}$ that reflect the temporal periodicity of the tidal magnetic field as well as the spatial localization of the sources within the oceans. An illustration for the M2 tide with $\omega = \frac{2\pi}{12.42h}$ can be found in Figure 3. For the computation of the $\tilde{\mathbf{B}}_\ell$ as solutions of (1), we have used the X3DG solver from Kuvshinov (2008).

Figure 4 shows the accumulated energy $\sum_{\ell=1}^{L} |\mathbf{u}_\ell(\xi)|^2$, for $\xi \in \mathbb{S}$, of the underlying functions $\mathbf{u}_\ell$ that describe the velocity field and the accumulated energy $\sum_{\ell=1}^{L} |\mathbf{B}_\ell^{\mathrm{re}}(x,t)|^2 + |\mathbf{B}_\ell^{\mathrm{im}}(x,t)|^2$, for $x \in \mathbb{S}_r$ with $r = a + 300\mathrm{km}$ and time $t = 0$, of the corresponding trial functions. In both cases, one can clearly see the spatial localization over the oceans. However, the accumulated energy of the trial functions additionally reflects the influence of the conductivity $\sigma$ and the main/core magnetic field $\mathbf{B}_{\mathrm{main}}$ indicated in Figure 3.

## 3 Examples

For our experiments we rely on the CM5 geomagnetic field model (cf. Sabaka et al. (2015)) and a forward model based on the M2 depth-integrated tidal velocity field from TPXO8-ATLAS[3] (cf. Egbert and Erofeeva (2002)) that has also been used in Kuvshinov (2008). The contribution of CM5 that is due to the oceanic M2 tide is given as an expansion in terms of spherical harmonics up to degree 18, we denote it as $\mathbf{B}_{\mathrm{oc}}^{\mathrm{CM5}}$ for the remainder of this section and sample it at $M = 250,000$ points which

---

[3]volkov.oce.orst.edu/tides/tpxo8_atlas.html

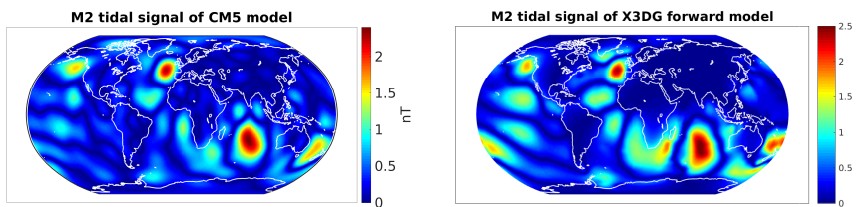

**Figure 5.** Absolute value of the radial part of the tidal model $\mathbf{B}_{\mathrm{oc}}^{\mathrm{CM5}}$ as well as the forward model $\mathbf{B}_{\mathrm{oc}}^{\mathrm{X3DG}}$ at an altitude of 300 km above the Earth's surface.

are taken from actual Swarm satellite tracks. The forward model has been computed via the X3DG solver based on the surface conductance and the main/core magnetic field model indicated in the bottom row of Figure 3 and a depth-integrated M2 tidal velocity field from TPXO8-ATLAS. We denote it by $\mathbf{B}_{\mathrm{oc}}^{\mathrm{X3DG}}$ and evaluate it on the same point grid as before. These samples are used as input data $\mathbf{b} \in \mathbb{R}^M$ for the Regularized (Orthogonal) Functional Matching Pursuit, which works iteratively as indicated

in (2). In the following, we want to illustrate the influence of the choice of different function systems (i.e., the choice of different dictionaries $\mathcal{D}$) on the approximation of $\mathbf{B}_{\mathrm{oc}}^{\mathrm{CM5}}$ and $\mathbf{B}_{\mathrm{oc}}^{\mathrm{X3DG}}$. For that purpose, we choose three different dictionaries: the spherical harmonic based

$$\mathcal{D}_1 = \{\cos(\omega t)\mathbf{h}_{n,k}(x), \sin(\omega t)\mathbf{h}_{n,k}(x)\}_{n=0,\dots,20,\,k=-n,\dots,n},$$

with $\omega = \frac{2\pi}{12.42h}$ and the functions $\mathbf{h}_{n,k}$ from (6); the Abel-Poisson kernel based

$$\mathcal{D}_2 = \{\cos(\omega t)\mathbf{k}(x, a\eta_i), \sin(\omega t)\mathbf{k}(x, a\eta_i)\}_{i=1,\dots,M_p},$$

where $a = 6371.2\text{km}$, $\{\eta_i\}_{i=1,\dots,M_p}$ is a Reuter grid on $\mathbb{S}$ with $M_p = 6201$ nearly equally distributed points (see, e.g., Michel (2013), p. 137), and $\mathbf{k}$ given as in (7); and

$$\mathcal{D}_3 = \{\mathbf{B}_{\ell}^{\mathrm{re}}(x,t), \mathbf{B}_{\ell}^{\mathrm{im}}(x,t)\}_{\ell=1,\dots,1200},$$

with $\mathbf{B}_{\ell}^{\mathrm{re}}$ and $\mathbf{B}_{\ell}^{\mathrm{im}}$ the physics based trial functions from (9) and (10).

The actual signals that we want to approximate are indicated in Figure 5. The approximations $\bar{\mathbf{B}}_N$ of $\mathbf{B}_{\mathrm{oc}}^{\mathrm{CM5}}$ together with the residuals $|\bar{\mathbf{B}}_N - \mathbf{B}_{\mathrm{oc}}^{\mathrm{CM5}}|$ for each of the three dictionaries above are shown in Figure 6. Whereas, the respective approximations of $\mathbf{B}_{\mathrm{oc}}^{\mathrm{X3DG}}$ and corresponding residuals $|\bar{\mathbf{B}}_N - \mathbf{B}_{\mathrm{oc}}^{\mathrm{X3DG}}|$ are displayed in Figure 7.

In the case of $\mathbf{B}_{\mathrm{oc}}^{\mathrm{CM5}}$ as the underlying signal, it can be seen in Figure 6 that the dictionary $\mathcal{D}_1$ yields the overall best approximation which, however, is not surprising since we try to fit a spherical harmonic based model with a spherical harmonic

based dictionary. The result for dictionary $\mathcal{D}_2$ shows a more localized pattern in the residual, as is expected for the use of Abel-Poisson kernels. However, the maxima in the residual are not correlated to specific continental or oceanic structures but they mainly coincide with the maxima of the original signal $\mathbf{B}_{\mathrm{oc}}^{\mathrm{CM5}}$. The situation for dictionary $\mathcal{D}_3$ of physics based trial functions is different. The agreement of the approximation with $\mathbf{B}_{\mathrm{oc}}^{\mathrm{CM5}}$ is good over the oceans but significant deviations exist

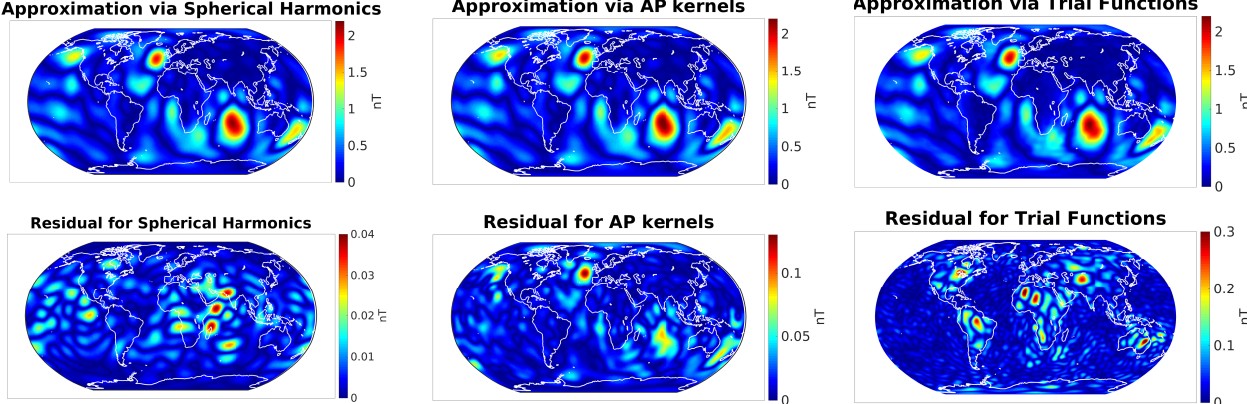

**Figure 6.** Absolute value of the radial part of approximations of $\mathbf{B}_{\mathrm{oc}}^{\mathrm{CM5}}$ based on dictionary $\mathcal{D}_1$ (top left), dictionary $\mathcal{D}_2$ (top center), and dictionary $\mathcal{D}_3$ (top right), as well as the corresponding residuals with respect to $\mathbf{B}_{\mathrm{oc}}^{\mathrm{CM5}}$ (bottom row). Note the different scales in the bottom row which are chosen in order to emphasize the spatial distribution of the residuals.

over the continents. The latter could be an indication that the original model $\mathbf{B}_{\mathrm{oc}}^{\mathrm{CM5}}$ contains contributions over the continents whose physical origin is not due to induction by oceanic tides. Some smaller deviations over oceanic areas exist around Southern Africa and East of Australia. Since we are dealing with the approximation of a low-degree (up to degree 18) spherical harmonic based M2 tidal magnetic field model by localized trial functions, one cannot reliably say if the latter deviations are

artifacts from the approximation procedure or if they have a physical origin. However, those are areas with a shallower ocean topography, so that the assumption of surface divergence-free depth integrated tidal velocities (which we made for our choice of the underlying $\mathbf{u}_\ell$) and the assumption of a constant ocean depth (we chose a depth of 1km for the generation of the $\tilde{\mathbf{B}}_\ell$ via the X3DG solver) might not be accurate in these areas. Nonetheless, the residuals over the continents show that the use of the adapted trial functions might eventually deliver improved tidal magnetic field models that correct unrealistic continental

contributions without disregarding continental areas entirely.

    The residuals of the approximations of the forward model $\mathbf{B}_{\mathrm{oc}}^{\mathrm{X3DG}}$ in Figure 7, on the other hand, indicate that the quality of the approximations does not vary too much (at least on scales that are relevant for satellite data approximation) among the three tested function systems. This is mainly due to the fact that the input model $\mathbf{B}_{\mathrm{oc}}^{\mathrm{X3DG}}$ already reflects certain spatial localization properties over the oceans. In such a scenario (if additionally solely interested in the approximation of the signal and not the

underlying velocity fields) it would, therefore, not be necessary to use the adapted trial functions that we have introduced. The crucial point, however, is that satellite data typically contains undesired contributions over the continents that are not due to ocean tide generated magnetic fields. In order to illustrate this behavior, we use the follwoing additional example. We take randomly distributed Fourier coefficients to construct a "noise" function $\mathbf{e}$ with a bandlimit of degree 40, i.e.

$$\mathbf{e} = \sum_{n=1}^{40} \sum_{k=-n}^{n} \hat{e}_{(n,k)} \mathbf{y}_{n,k}^{(3)},$$

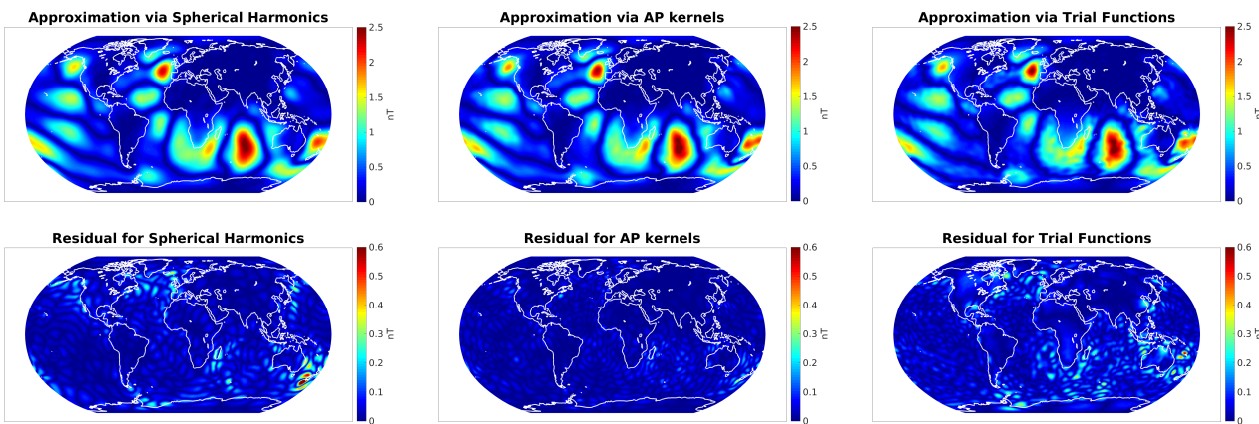

**Figure 7.** Absolute value of the radial part of approximations of $\mathbf{B}_{\mathrm{oc}}^{\mathrm{X3DG}}$ based on dictionary $\mathcal{D}_1$ (top left), dictionary $\mathcal{D}_2$ (top center), and dictionary $\mathcal{D}_3$ (top right), as well as the corresponding residuals with respect to $\mathbf{B}_{\mathrm{oc}}^{\mathrm{X3DG}}$ (bottom row).



**Figure 8.** Absolute value of the radial part of $\mathbf{B}_{\mathrm{oc}}^{\mathrm{X3DG}}$ (left), model of continental 'noise' $\mathbf{e}$ (center) as well as superposition $\mathbf{B}_{\mathrm{oc}}^{\mathbf{e}}$ of both of the latter (right).

where the Fourier coefficients $\hat{e}_{(n,k)}$ are normally distributed with zero mean and a variance such that the amplitude of $\mathbf{e}$ is in the range of the oceanic signal $\mathbf{B}_{\mathrm{oc}}^{\mathrm{X3DG}}$. This function $\mathbf{e}$ is then restricted to the continents and eventually superposed with the forward model (cf. Figure 8). For the sake of clarity, we denote the approximations of the noisy data

$$\mathbf{B}_{\mathrm{oc}}^{\mathbf{e}} = \mathbf{B}_{\mathrm{oc}}^{\mathrm{X3DG}} + \mathbf{e}$$

5    by $\bar{\mathbf{B}}_N^{\mathbf{e}}$ instead of $\bar{\mathbf{B}}_N$. The latter still represents the approximation of $\mathbf{B}_{\mathrm{oc}}^{\mathrm{X3DG}}$ without extra continental noise $\mathbf{e}$.

In Figure 9 one can directly see the influence which the continental noise has on the approximation depending on the various dictionaries (Figure 7 shows the same quantities for the approximations in the undisturbed setup). In the case of dictionary $\mathcal{D}_1$, the spherical harmonics also approximate a part of the continental data which in turn also has some impact on the approximation in oceanic areas. Due to the localization of the kernels contained in dictionary $\mathcal{D}_2$, the (undesired)

10    reconstruction of the continental noise is even more accurate while the reconstruction over the oceans only changes very slightly. With the proposed physics based functions in dictionary $\mathcal{D}_3$, however, the influence of the continental noise is much less apparent. The maxima occur very close to the coast lines, which is most likely due to numerical issues stemming from discontinueties of the data $\mathbf{B}_{\mathrm{oc}}^{\mathbf{e}}$ in coastal areas. A closer look at the differences $|\bar{\mathbf{B}}_N - \bar{\mathbf{B}}_N^{\mathbf{e}}|$ between the approximations of

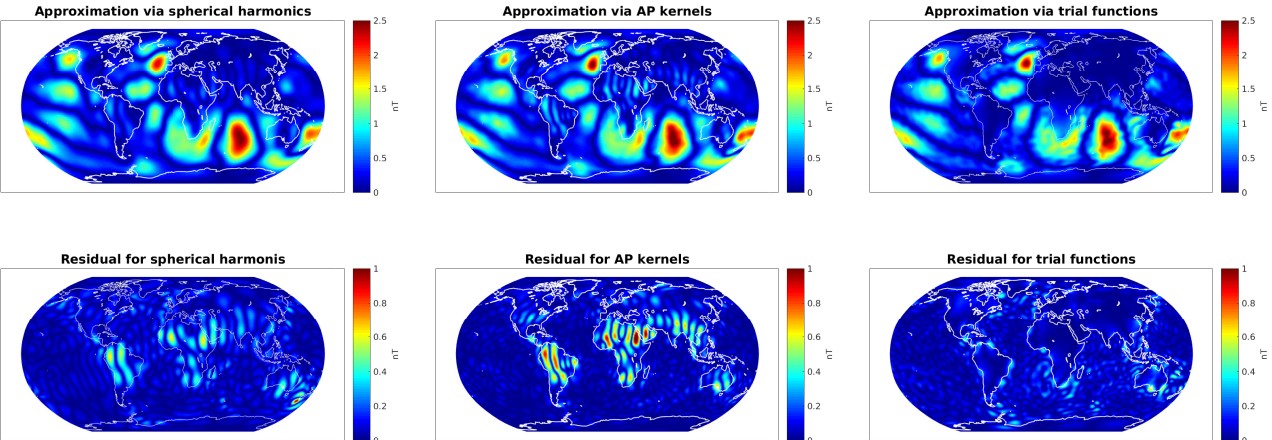

**Figure 9.** Absolute value of the radial part of approximations of superposition $\mathbf{B}_{\text{oc}}^{\mathbf{e}}$ based on dictionary $\mathcal{D}_1$ (top left), dictionary $\mathcal{D}_2$ (top center), and dictionary $\mathcal{D}_3$ (top right), as well as the corresponding residuals with respect to original $\mathbf{B}_{\text{oc}}^{\text{X3DG}}$ without the continental 'noise' $\mathbf{e}$ (bottom row).



**Figure 10.** The difference between the approximations $\bar{\mathbf{B}}_N$ of the undisturbed $\mathbf{B}_{\text{oc}}^{\text{X3DG}}$ (as shown in the top row of Figure 7) and the approximations $\bar{\mathbf{B}}_N^{\mathbf{e}}$ of the noisy $\mathbf{B}_{\text{oc}}^{\mathbf{e}}$ (as shown in the top row of Figure 9).

noisy and undisturbed data is given in Figure 10. This shows again that the inclusion of continental noise has a smaller effect on the approximation via physics-based trial functions than on the approximations via the other tested trial functions. Moreover, the corresponding root mean square errors of the approximations $\bar{\mathbf{B}}_N$ and $\bar{\mathbf{B}}_N^{\mathbf{e}}$, respectively, can be found in Table 1. In both cases, we compared the approximations to the undisturbed data $\mathbf{B}_{\text{oc}}^{\text{X3DG}}$ in order to emphasize the impact of continental noise on the overall approximation. The errors over continental and oceanic regions are provided seperately.

## 4 Conclusions

The main goal of this paper is to study the errors that are made by the approximation of tidal magnetic fields by use of different sets of trial functions. While, e.g., Saynisch et al. (2018) compared forward models for the M2 tidal magnetic field based on different tidal models, we aim at illustrating the effect of the involved trial functions on the possible extraction of the tidal

| RMS | $\lvert\bar{\mathbf{B}}_N - \mathbf{B}_{\mathrm{oc}}^{\mathrm{X3DG}}\rvert$ | | | $\lvert\bar{\mathbf{B}}_N^{\mathbf{e}} - \mathbf{B}_{\mathrm{oc}}^{\mathrm{X3DG}}\rvert$ | | | $\lvert\bar{\mathbf{B}}_N - \bar{\mathbf{B}}_N^{\mathbf{e}}\rvert$ | | |
|---|---|---|---|---|---|---|---|---|---|
| | $\mathcal{D}_1$ | $\mathcal{D}_2$ | $\mathcal{D}_3$ | $\mathcal{D}_1$ | $\mathcal{D}_2$ | $\mathcal{D}_3$ | $\mathcal{D}_1$ | $\mathcal{D}_2$ | $\mathcal{D}_3$ |
| overall | 0.060533 | 0.029952 | 0.064133 | 0.107972 | 0.128185 | 0.086386 | 0.089407 | 0.126170 | 0.057929 |
| continents | 0.045495 | 0.021553 | 0.055487 | 0.160529 | 0.237245 | 0.085791 | 0.153771 | 0.236059 | 0.067152 |
| oceans | 0.064366 | 0.032047 | 0.086563 | 0.086145 | 0.066827 | 0.086563 | 0.057396 | 0.062991 | 0.054858 |

**Table 1.** Root mean square (RMS) errors corresponding to the approximations of the undisturbed foward model $\mathbf{B}_{\mathrm{oc}}^{\mathrm{X3DG}}$ as well as the noisy model $\mathbf{B}_{\mathrm{oc}}^{\mathbf{e}} = \mathbf{B}_{\mathrm{oc}}^{\mathrm{X3DG}} + \mathbf{e}$ compared to the 'ground truth' $\mathbf{B}_{\mathrm{oc}}^{\mathrm{X3DG}}$ with the three different dictionaries. Left-hand columns show errors for the approximation of $\mathbf{B}_{\mathrm{oc}}^{\mathrm{X3DG}}$ (compare Figure 7), while center columns show errors for the approximation of $\mathbf{B}_{\mathrm{oc}}^{\mathbf{e}}$ (compare Figure 9). The right-hand columns display the RMS errors of the difference between the respective approximations (as shown in Figure 10).

magnetic field from satellite data in the first place. The indicated residuals for the synthetic examples show that the use of the presented adapted physics based trial functions could have a detectable effect for the extraction of such signals in satellite data. These trial functions reflect the underlying physics in the sense that they satisfy the time harmonic Maxwell equations and that they include knowledge of the ambient core magnetic field and the Earth's conductivity, but they are not designed to rely on detailed oceanographic models. The latter can be advantageous since the extracted magnetic field induced by ocean tides might eventually be used to make inferences on such models.

*Competing interests.* None

*Acknowledgements.* This work was supported by DFG grant GE 2781/1-1 within the Reserach Priority Program SPP 1788 "DynamicEarth". The authors thank Alexey Kuvshinov and Alain Plattner for sharing their codes for the X3DG solver and for the generation of vectorial Slepian functions, respectively.

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
