# Peer review of "On the Approximation of Spatial Structures of Global Tidal Magnetic Field Models"

_Annales Geophysicae, 2018_

## Referee Comment (RC1) · Anonymous Referee #1 · 16 Jul 2018

**General comments:**

The paper describes a new approach for the extraction of (M2) tidally induced magnetic fields from satellite magnetometer observations. The proposed method uses spatial constrains in addition to the usually used frequency constrains during the M2 extraction. Variants of the new method and the traditional approach are discussed and compared. The paper is well written and the topic itself is very interesting. It is about time to go beyond the traditional tidal extraction methods by including additional information. Successful application of the described methods to real observational data would be valuable to the community. The presentation of the results has to be improved.

[Figure]

**Specific comments:**

As for now, the authors demonstrate (by hard to read figures) that their method can give comparable results to the traditional (spherical harmonic) approach. At least over the ocean. At least the figures should be improved and enforced with some numbers, e.g., the performance of the approaches are indistinguishable over the ocean (fig. 67, see also technical corrections).

Consequently, the paper would improve greatly if the authors could demonstrate the main purpose of their physical trial-functions: to separate between oceanic contributions and artifacts from land. The authors claim several times that their method could be used to remove undesired terrestrial contributions. I would recommend to add such contributions to their observations ($B_{oc}^{CM5}, B_{oc}^{X3DG}$) and (re)apply their method to prove that claim.

**Technical corrections:**

Since this is a physical journal it would help to point out mathematical defined terms that are not so common in this community: e.g., trial function, greedy algorithm, foundation (=basis?), dictionary and so on. Either ad a "so called" in front of them, use italic font or apostrophes.

Signals on the continents should be clearly distinguishable from oceanic contributions: Please add visible coastal lines to all plots and unify the plotted range where applicable.

Fig. 6-7: It would make sense to plot also the differences of the residuals. For example, keep the D1 residuals and plot the D2-D1 and D3-D1 residuals. The influence of the dictionaries on the oceanic residuals is not visible in the plots! At least give numbers (rms) for the residuals from all dictionaries and discuss them in the text. It would be a good idea to calculate separate rms for land and ocean. Please, unify the scales(!)

and add visible coastlines.

page 2, line 2: add reference for "greedy algorithms" page 7: remove eq. 9 page 7, line 16: how is 'best localized' determined? Based on the eigenvalue's proximity to 1? page 8, line 17: add reference for TPXO.

---

## Referee Comment (RC2) · Anonymous Referee #2 · 16 Aug 2018

GENERAL COMMENTS:

This paper discusses the construction and use of customized basis functions for the purpose of modeling or representing global tidal magnetic fields. These functions include the temporal periodicities, but also take into account the spatial patterns of the tides by confining their variation within ocean basins. Three types of functions are considered, including traditional solid harmonics, radial basis functions, and physically motivated functions that solve Maxwell's equations, but are restricted to ocean basins via Slepian functions. They apply these to the CM5 M2 field as well as a synthetic field and find that there are discrepancies between the CM5 model and the physics-based functions over the continents. The conclusion is that these physical functions could be a more appropriate set of basis functions for modeling tidal magnetic fields. I believe

the paper is acceptable for publication in the present form after some minor revisions listed below, but it would be much improved after addressing the major concerns I discuss below.

SPECIFIC COMMENTS:

I find the paper to be technically very good, but the reduction of noise over the continents does not seem so important. CM5 was based upon CHAMP data only, but newer M2 models from Swarm (see Sabaka et al., 2018) show much less noise contamination in general, which will probably also be reduced over continents. The M2 signal is small and the models are not usually interpreted over the continents anyways, so this problem is even more secondary. In fact, the residuals for the trial functions in figure 6 (CM5) are over a smaller range of 0.0-0.3 than those of the figure 7 (synthetic) range 0.0-0.6, which makes me wonder if the appearance of the residuals for CM5 look more exaggerated than they really are.

However, what I find very important about this work are the physical based trial functions themselves because they allow a direct connection between the data and the ocean velocity and conductivity parameters. I think the most important part of the paper would be an added section that addresses solving the inverse problem of inferring these parameters from the data, that is, CM5, using these trial functions. The section should include discussions on:

1) How does the u field corresponding to figure 6 from CM5 compare with the TPXO8-ATLAS model?

2) Can you solve for the 1D conductivity sigma in the ball $B_a$ like was done in Grayver et al. (2016)? Even if you cannot do this at least discuss the issues like sensitivity, observability, regularization, etc.

3) By fixing u to TPXO8-ATLAS, can you say anything about the sensitivity or observability of sigma in the shell $S_a$?

If this discussion is added, then I think the paper would be much more useful.

TECHNICAL CORRECTIONS:

1) Define E and \mu_0 in equation 1.

2) Perhaps you could give some more detail about the Regularized Orthogonal Functional Matching Pursuit algorithm.

3) What is the meaning of the norm with respect to H in equation 2?

4) Should d have a subscripted i or i-1 in equation 2?

5) Define \lambda in equation 2.

6) On page 4 you say y_{0,0}^(2) = y_{0,0}^(3) = 0 for convenience, but the real reason is to ensure uniqueness?

7) In the context of eigenvalue equation (not numbered) on page 6, you should state that the denominator of equation 8 is equal to \hat{g}^T \hat{g}. If this is not correct, then please explain the eigenvalue equation more thoroughly.

8) What model does B_{main} come from?

9) What are the units of the map in figure 4a?

10) On page 9 you define D_1 from n = 0,...,20, but if these functions are the solid harmonics, then should you be starting at n = 1 instead?

11) Define a Reuter grid or give a reference.

12) What altitude are the maps in figure 5?

13) In all global map figures you should outline the continents so they can be seen.

14) Figures 6 and 7 should be combined such that each corresponding map uses the same scale and can be seen simultaneously. As stated earlier, it appears that the residuals for the CM5 fits are over a smaller range, which amplifies the features relative
to that of X3DG.

---

## Author Comment (AC1) · 18 Sep 2018

We thank the anonymous reviewer for the valuable suggestions.

**Concerning the specific comments:**

- *"At least the figures should be improved and enforced with some numbers, e.g., the performance of the approaches are indistinguishable over the ocean . . . ."*

  We have added Table 1 at the end of Section 3 that includes the root mean square error (rms) of all presented approximations, together with some further explanation within the main text. The root mean square error is indicated separately for oceans and continents.

- *". . . the paper would improve greatly if the authors could demonstrate the main purpose of their physical trial-functions: to separate between oceanic contributions and artifacts from land. The authors claim several times that their method could be used to remove undesired terrestrial contributions. I would recommend to add such contributions to their observations ($\mathbf{B}_{oc}^{CM5}$ , $\mathbf{B}_{oc}^{X3DG}$) and (re)apply their method to prove that claim.."*

  An additional example (Figures 8,9,10) has been added at the end of Section 3 that addresses these issues. There, we compare the approximations of a M2 tidal magnetic field model with and without artificial 'noise' over the continents.

**Concerning the technical corrections:**

- *"Since this is a physical journal it would help to point out mathematical defined terms that are not so common in this community"*

  Mathematical terms that are not clear to a larger audience have been reformulated or explained:

  - p.2, l.14: "... serves as a foundation for the approximation ..." has been reformulated to "... is suitable for the approximation ..."
  - The term "greedy-type algorithm" has been deleted as it does not have any particular importance for the paper.
  - The terms "dictionary" and "trial functions" are explained after their first appearance on p.2, l.30 and p.3, l.8.

- *"Signals on the continents should be clearly distinguishable from oceanic contributions: Please add visible coastal lines to all plots and unify the plotted range where applicable."*

  Coastlines have already been included in the original images, but they have been made clearer in the new versions of Figures 5,6,7.

- *"Fig. 6-7: It would make sense to plot also the differences of the residuals. For example, keep the D1 residuals and plot the D2-D1 and D3-D1 residuals. The influence of the dictionaries on the oceanic residuals is not visible in the plots! At least give numbers (rms) for the residuals from all dictionaries and discuss them in the text. It would be a good idea to calculate separate rms for land and ocean."*

  We decided not to include further differences of residuals D3-D1 and D2-D1 for Figures 6,7. In particular, the residuals for the different dictionaries are of different orders of magnitude (in Figure 6, e.g., the residual for spherical harmonics has a maximum of

0.04 while the residual for the physics-based trial functions has a maximum of 0.3), so that the differences would not yield any visible additional information. However, we adjusted the scales in Figure 6 to improve the representation of the spatial distribution of the residuals.

Additionally, for the new example with artificial 'noise' over the continents, we have added a figure indicating the differences between the approximation of a signal with and without continental 'noise' (cf. Figure 10). We hope that this way the influence of the dictionaries on the oceanic and continental residuals becomes clearer.

- *"Please, unify the scales(!)"*

  We now have used unified scales within sets of pictures containing the same information. An exception where we kept non-unified scales are the residuals in Figure 6. There, the goal was to highlight the spatial distribution of the differences rather than their amplitudes. This, we think, is better achieved by non-unified scales.

- *"page 7: remove eq. 9 page 7"*

  Equation (9) has been removed.

- *"line 16: how is best localized determined? Based on the eigenvalues proximity to 1?"*

  "best localized" in terms of Slepian functions means an energy ratio (i.e., the eigenvalue of a Slepian function) that is close to one, as indicated by the reviewer. This we now have mentioned explicitly after the first appearance of "best localized" before equation (8).

- *"page 8, line 17: add reference for TPXO"*

  A proper reference for TPXO has been added.

---

## Author Comment (AC2) · 18 Sep 2018

We thank the anonymous reviewer for the valuable suggestions.

**Concerning the specific comments:**

- *"I find the paper to be technically very good, but the reduction of noise over the continents does not seem so important. CM5 was based upon CHAMP data only, but newer M2 models from Swarm (see Sabaka et al., 2018) show much less noise contamination in general, which will probably also be reduced over continents. ... In fact, the residuals for the trial functions in figure 6 (CM5) are over a smaller range of 0.0-0.3 than those of the figure 7 (synthetic) range 0.0-0.6, which makes me wonder if the appearance of the residuals for CM5 look more exaggerated than they really are. "*

  It is true that the scales of Figures 6 and 7 are different. However, this is to illustrate different aspects of the approximations via the different trial functions, as we explain in a bit more detail in our reply to the last question in the technical corrections. It was not meant to exaggerate the residual of one magnetic field model over the other.

- *"However, what I find very important about this work are the physical based trial functions themselves because they allow a direct connection between the data and the ocean velocity and conductivity parameters. I think the most important part of the paper would be an added section that addresses solving the inverse problem of inferring these parameters from the data, that is, CM5, using these trial functions. The section should include discussions on:*
  *1) How does the* **u** *field corresponding to figure 6 from CM5 compare with the TPXO8-ATLAS model?*
  *2) Can you solve for the 1D conductivity sigma in the ball* $\mathbb{B}_a$ *like was done in Grayver et al. (2016)? Even if you cannot do this at least discuss the issues like sensitivity, observability, regularization, etc.*
  *3) By fixing u to TPXO8-ATLAS, can you say anything about the sensitivity or observability of sigma in the shell* $\mathbb{S}_a$*?"*

  As the reviewer suggests, the crucial aspect is the connection between the oceanic magnetic signal and the underlying quantities. In particular, **u** can be obtained directly from the approximation of $\mathbf{B}_{oc}$ via the proposed trial functions, due to the linear connection. A comparison with TPXO8-ATLAS has not been included in the paper since we based our trial functions solely on divergence-free (depth-integrated) current systems, and which we assume has lead to reconstructed current systems that are not comparable to TPXO8-ATLAS. However, the inclusion of trial functions also based on non-divergence-free current systems and a thorough study and comparison to existing velocity field models will be part of future work. It should be said, though, that one cannot expect the resolution of, e.g., TPXO8-ATLAS solely based on magnetic field satellite data.

  Concerning questions 2) and 3): due to the nonlinear connection between the magnetic field and the underlying conductivity, it is significantly more difficult to reconstruct $\sigma$ (assuming a fixed **u**) than reconstructing **u** (assuming a fixed $\sigma$). Of course, one could base our trial functions on different conductivity models and then compare the approximations of $\mathbf{B}_{oc}$. However, our approach does not provide a nice and simple possibility of reconstructing $\sigma$ simultaneously together with $\mathbf{B}_{oc}$. Therefore, at the moment, we focus on the simultaneous reconstruction of **u** and $\mathbf{B}_{oc}$.

These issues are now briefly addressed in the penultimate paragraph of the introduction of the paper.

**Concerning the technical corrections:**

- *"Define $\mathbf{E}$ and $\mu_0$ in Equation (1)."*

  We added the mentioned definitions.

- *"Perhaps you could give some more detail about the Regularized Orthogonal Functional Matching Pursuit."*

  The functional matching pursuit is essentially a method that iteratively picks functions from a pre-defined set of trial functions ("dictionary") for a given minimization problem (in our case, data misfit plus a regularization term). Since the minimization algorithm is not the crucial aspect of this paper (we chose the functional matching pursuit due to its flexibility but it is not necessary to use that algorithm for the proposed trial functions), we desisted from a further explanation of the algorithm. For a more detailed explanation of the algorithm, we supplied some references in the beginning of Section 2.

- *"What is the meaning of the norm with respect to $\mathcal{H}$ in Equation (2)?"*

  We added an according footnote.

- *"Should $d$ have a subscripted $i$ or $i-1$ in Equation (2)?"*

  No, we minimize over all $d \in \mathcal{D}$ in order to find the next $d_i$. The function $d_{i-1}$ is implicitly contained in both $\mathbf{B}_{i-1}$ and $\mathbf{R}_{i-1}$.

- *"Define $\lambda$ Equation (2)."*

  An explanation has been added.

- *"On page 4 you say $y_{0,0}^{(2)} = \mathbf{y}_{0,0}^{(3)} = \mathbf{0}$ for convenience, but the real reason is to ensure uniqueness?"*

  This is in fact only for convenience in order to be able to let all sums start with $n = 0$ since neither $\mathbf{y}_{n,k}^{(2)}$ nor $\mathbf{y}_{n,k}^{(3)}$ are actually defined for $n = 0$ because the denominator becomes zero in Equations (4) and (5).

- *"In the context of eigenvalue equation (not numbered) on page 6, you should state that the denominator of equation 8 is equal to $\hat{g}^T \hat{g}$. If this is not correct, then please explain the eigenvalue equation more thoroughly. "*

  Indeed, in case of a normalized function the denominator is equal to $\hat{g}^T \hat{g}$. We described this in more detail now.

- *"What model does $\mathbf{B}_{\mathrm{main}}$ come from?"*

  We used the main field from CHAOS-5, which we now also cited.

- *"What are the units of the map in figure 4a?"*

  The unit in Figure 4a would be $m^2/s^2$ and in Figure 4b it would be $nT^2$. However, in the revision, to avoid confusion, we deleted the units both for Figure 4a and 4b since the accumulated energy is no actual 'physical' energy.

- *"On page 9 you define $\mathcal{D}_1$ from $n = 0, \ldots, 20$, but if these functions are the solid harmonics, then should you be starting at $n = 1$ instead? "*

  We also incorporate the case $n = 0$ which is $\mathbf{h}_{0,0}(r\xi) = \frac{1}{r^2}\xi Y_{0,0} = \frac{1}{r^2\sqrt{4\pi}}\xi$.

- *"Define a Reuter grid or give a reference."*

  We added a reference.

- *"What altitude are the maps in figure 5?"*

  We rectified the corresponding caption and added the 300km altitude.

- *"In all global map figures you should outline the continents so they can be seen."*

  We improved the visibility of the coast lines.

- *"Figures 6 and 7 should be combined such that each corresponding map uses the same scale and can be seen simultaneously. As stated earlier, it appears that the residuals for the CM5 fits are over a smaller range, which amplifies the features relative to that of X3DG."*

  We chose different ways to display the CM5 and the X3DG residuals in order to emphasize certain aspects of the approximation. In Figure 6, where CM5 data is approximated, we chose not to unify the scales in order to improve the display the different spatial structures of the residuals. That is, spherical harmonic create global errors, while the approximation with kernels results in localized errors, and the proposed new trial functions lead to larger residuals over the continents. A unification of the scales would make those differences less visible. In particular, the global structures of the polynomial spherical harmonic solution would hardly be visible in an 0.0 to 0.3nT scale. The goal of Figures 6 and 7 was not to compare the quality of CM5 and X3DG but to illustrate the effect of the choice of different trial functions (since the numerical computation of the physics-based trial functions is based on the X3DG solver, it is not surprising that the residuals over the continents seem better for the approximation of X3DG than for CM5).

  Moreover, we added another (new) example in order to more thoroughly discuss the influence of continental 'noise' in the data (Figures 8,9,10). Here, we compare the residuals of approximations from data with and without continental 'noise' and we also compare corresponding RMS errors in a separate table.